# Series Arc Fault Detection under Vibration Condition Based on N-M-M-B

**DOI:** 10.3390/s24030959

**Published:** 2024-02-01

**Authors:** Yanli Liu, Ganqing Yang, Huiyang Wang

**Affiliations:** Department of Electrical and Control Engineering, Liaoning Technical University, Huludao 125105, China

**Keywords:** series arc fault, mechanical fault of the motor, noise-assisted multivariate empirical mode decomposition, multi-scale convolutional neural network, bidirectional long short-term memory network

## Abstract

Under the conditions of a mechanical fault in a motor, mechanical vibration of a specific frequency can be generated. The electrical contact points directly connected to the motor can vibrate at the same frequency. The electrical contact points with poor contact can easily produce a series arc fault under vibration conditions, which affects the reliability of the power supply. In order to detect the series arc fault under different vibration conditions, the arc fault generator is connected between the back end of the frequency converter and the motor. An arc fault experiment under different vibration conditions was carried out and the fault phase current and arc voltage signals were collected. In this paper, the noise-assisted multivariate empirical mode decomposition and the correlation coefficient between each intrinsic mode function are used to select the fault feature signals. Then, the reconstructed signal is input into the series arc fault model combining a multi-scale convolutional neural network and a bidirectional long short-term memory network for training. The research results show that the series arc fault detection method proposed in this paper can effectively detect the series arc fault and can preliminarily identify the type of motor fault causing the mechanical vibration of the motor; the model has good noise immunity and generalization.

## 1. Introduction

Due to the poor contact and corrosion of contact points, a series arc fault (SAF) may occur in the frequency converter loop of an industrial motor [1]. When mechanical faults such as rotor imbalances, air gap dynamic eccentricities, or the poor processing and assembly of rolling bearings, occur in the motor, a mechanical vibration of a specific frequency will be generated. The mechanical vibration will aggravate the SAF at the electrical contact point with poor contact. Whether there is a mechanical vibration fault in the motor, whether the arc occurs in the front end of the frequency converter or the back end of the frequency converter, and whether the motor is running with the frequency converter will affect the signal features of the SAF. In addition, the SAF itself has concealment, randomness and complexity, which will aggravate the difficulty of arc fault detection under mechanical vibration conditions, resulting in the SAF becoming one of the most difficult types of line faults to detect [2]. Therefore, based on the loop current, the arc fault detection model of the motor frequency converter load under vibration conditions is established for judging whether the arc fault occurs and for preliminarily judging the type of motor fault, which is of a certain significance for improving the power supply safety performance of an industrial motor frequency converter loop.

Regarding the aspect of feature extraction of an arc fault current and voltage signals, Gustavo S. da Rocha [3] took the loop current as the analysis object, and combined wavelet decomposition with the support vector machine (SVM) to realize the detection of the SAF. Yu [4] used improved complete ensemble empirical mode decomposition with adaptive noise (ICEEMDAN) to decompose the arc fault current data, and then constructed the detection variables to distinguish the arc fault from the normal state by selecting the intrinsic mode function (IMF) and calculating its variance. Ali Amiri [5] carried out recurrence quantification analysis (RQA) on the original current, and used the definition of the determinism parameter (DET) as the characteristic quantity to realize the SAF identification of a photovoltaic system using the threshold method. G. Zou [6] obtained the arc fault current component by the complete ensemble empirical mode decomposition with adaptive noise (CEEMDAN), and proposed a highly identifiable feature mining method based on the maximal information coefficient and the feature change significance. Z. Wang [7] proposed an SAF identification method based on the variational mode decomposition (VMD) and support vector machine. Although the feature extraction of arc fault data has been carried out in References [3,4,5,6,7], there are still some problems, such as the modal aliasing or average error of the set in the data processed by wavelet decomposition, CEEMDAN, RQA and other algorithms.

Because deep learning can mine the hidden features in the signal, it has been widely used in the field of SAF detection in recent years. Jiang [8] extracted the features of arc current signals through time domain, frequency domain and time-frequency domain analysis, and constructed a comprehensive arc fault detection model with load type detection and position determination function the combining random forest (RF) with deep neural networks (DNNs). Gao [9] used the squeeze-excitation network-deep convolution generative adversarial network (SE-DCGAN) to enhance the Gramian angular summation fields (GASF) image of the arc fault data to solve the problem of limited SAF samples. Liu [10] extracted the Hurst exponent, inter-harmonic variance and wavelet energy entropy (H-I-W) as the three-dimensional feature of the arc, and constructed the arc fault diagnosis model by combining the convolutional neural network (CNN) and the long short-term memory network (LSTM), which enhanced the fault identification ability. Xin [11] proposed a lightweight arc fault detection method based on an EffNet module, which can reduce the complexity of the algorithm at the same detection accuracy level. Although the method based on artificial intelligence has achieved good arc fault detection accuracy, it is not clear whether the results of References [8,9,10,11] are applicable to the arc fault detection of a motor frequency converter load.

In the field of industrial system arc fault diagnosis, W. Li [12] carried out the SAF experiment in a multi-load system. Through the construction, training and testing of the recurrent neural network (RNN), the diagnosis and line selection of the arc fault were realized. C. Han [13] proposed an arc fault identification method based on the kernel principal component analysis (KPCA) and firefly algorithm optimized support vector machine (FA-SVM), and verified the effectiveness of the method under complex harmonic conditions. H. Gao [14] established a low-dimensional arc fault feature vector by using the fractional Fourier transform (FRFT) and SVM, and proposed an arc fault detection and phase selection model. In References [12,13,14], the problem of arc fault detection under vibration conditions was not discussed, and the influence of the mechanical fault of the motor itself on the arc fault was not considered.

In References [3,4,5,6,7,8,9,10,11,12,13,14], good results have been achieved in the aspects of arc fault signal preprocessing, feature analysis and detection model establishment, but there are still the following shortcomings. In References [3,4,5,6,7], when the fault data are preprocessed, the empirical mode decomposition (EMD) correlation algorithms have problems such as modal aliasing or difficulty in aligning IMF components, which will have a certain impact on the diagnosis results. In References [8,9,10,11], the deep learning algorithm performs well in arc fault detection, but there is still some room for optimization. In Reference [10], only the global features extracted by the hidden layer at the end of CNN module are transmitted between the long short-term memory (LSTM) module and CNN module, and the local features contained in the middle hidden layer of CNN are not included. As a result, the global features extracted by the constructed neural network only contain high-level features such as sequence information, and it is difficult to describe low-level features such as single-peak signals. In References [12,13,14], the problem of arc fault detection of a motor frequency converter load under no vibration conditions was studied. But it is not clear what the characteristics are of the SAF of electrical contact points under vibration conditions and whether the cause of a motor fault can be preliminarily analyzed via the loop current.

To address the above problems, an arc fault generator with adjustable vibration frequency and amplitude is designed in this paper, which can simulate the SAF under different vibration conditions. The noise-assisted multivariate empirical mode decomposition (NA-MEMD) arc feature extraction method is constructed by combining the noise channel signal and the multivariate empirical mode decomposition, so as to better extract the frequency domain characteristics of the SAF current under vibration conditions and solve the problem of feature extraction modal aliasing. Meanwhile, the arc voltage and circuit are simultaneously analyzed to avoid misjudgment caused by single information. Finally, multi-scale technology is combined with the traditional CNN and bidirectional long short-term memory network (BiLSTM) to construct the MSCNN-BiLSTM arc fault detection model, so that the BiLSTM module can receive more and more abundant multi-level features from the CNN and improve the accuracy of the detection model.

## 2. Arc Fault Data Feature Extraction

### 2.1. Principle Analysis of Feature Extraction Algorithm

The time series of arc current and voltage signals are nonlinear and non-stationary signals. The EMD has excellent performance in dealing with such signals, but it has the problem of uncertainty of scale arrangement when dealing with multivariate signals. Therefore, in order to realize the simultaneous processing of current and voltage time series, this paper proposes the NA-MEMD algorithm. This algorithm can simultaneously decompose the voltage and current data, reduce the time of data feature extraction and avoid the modal aliasing phenomenon of the EMD.

#### 2.1.1. Multi-Experience Mode Decomposition (MEMD)

In order to solve the problem that the local maximum and minimum values in multivariate signals are not directly defined, the MEMD is used to decompose multi-channel signals. Firstly, the spatial multi-dimensional direction vector is established, the multi-dimensional signal is, respectively, projected to each direction of the space and the projection envelope and its mean are obtained. Finally, the multivariate IMF (MIMF) is obtained by subtracting the mean from the signals of each dimension. The MEMD flow chart is shown in Figure 1.

First, the Hammersley sequence sampling method is used to obtain a uniform sampling point set on the (n − 1)-dimensional sphere, and the direction vector of the n-dimension space is obtained. After that, the mapping of the input signal $h_{p-1}(t)$ in each direction of the vector $D^{\theta_{k}}$ is calculated and the data poles are interpolated. According to the center of gravity of the extreme point, the mean value of the extreme envelope and the interpolation of the input signal are calculated. Finally, the size of the decomposition margin and the given threshold is judged. If the margin is less than a given threshold or a monotone vector, the algorithm ends and all the MIMF and the remaining components are obtained. Otherwise, let i=i+1 and repeat the above steps. Finally, the signal ${\left\{X(t)\right\}}_{t=1}^{T}$ to be decomposed is expressed as follows:(1)$${\left\{X(t)\right\}}_{t=1}^{T}=\sum_{m=1}^{M}c_{m}(t)+r_{M}(t)$$
where $c_{m}(t)(1⩽m⩽M)$ is the extracted M MIMFs and $r_{M}(t)$ is the remaining residue.

#### 2.1.2. Noise-Assisted Multivariate Empirical Mode Decomposition (NA-MEMD)

By introducing additional noise channel signals and using the filter bank characteristics shown by white noise, the modal aliasing and mode alignment problems in MIMF are avoided. The core of the NA-MEMD algorithm is to splice the v-dimension independent white noise into the input signal of the s-dimension to form a signal to be decomposed with the (s + v)-dimension. Then, the MEMD method is used to process the corresponding MIMF component, from which the white noise is discarded. The MIMF component of the original signal is obtained [15,16,17]. The steps of the NA-MEMD algorithm are shown in Figure 2.

The NA-MEMD algorithm is a multivariate noise expansion form of the EMD. The algorithm not only makes full use of the fixed passband frequency characteristics of the MEMD when dealing with white noise, but also adds additional independent white noise to ensure that the IMF components of the decomposed signal and noise are completely separable [18,19]. Compared with the decomposition method based on EEMD, NA-MEMD does not need to perform the ensemble average of IMF, which improves the computational efficiency, reduces the noise interference and produces a better performance.

## 3. Arc Fault Detection Model

Because the deep learning network can mine the hidden features of signals and has strong learning ability, it is favored by the majority of fault detection researchers. The arc fault detection model is established based on MSCNN-BiLSTM. It solves the problem that the CNN has difficulties in describing the low-level features such as the single peak of signal and the problem that a single LSTM structure model is not excellent enough in feature extraction efficiency and performance. The MSCNN-BiLSTM network structure is shown in Figure 3. After connecting and flattening the output of the last two convolutional layers in CNN with the output of the pooling layer, it is inputted into BiLSTM to realize the detection of the SAF. Compared with CNN-LSTM, MSCNN-BiLSTM can make full use of the different levels of degradation features extracted from CNN to improve the accuracy of the model.

### 3.1. Multi-Scale Convolutional Neural Network (MSCNN)

The CNN is a common deep learning network model in the field of fault detection. It is inspired by the biological natural visual cognitive mechanism [20,21,22]. The data are analyzed by local link and weight sharing. On the one hand, this reduces the number of weights and makes the network easy to optimize. On the other hand, it also greatly reduces the complexity of the model and reduces the probability of over-fitting. Aiming at the problem that the CNN has difficulties in describing low-level features such as the single peak signal, this paper proposes the MSCNN based on arc fault detection. MSCNN takes the output features of the last layer as global features, and the output features of the middle layer as local features. The output features of a certain middle layer or several middle layers are connected and flattened with the output features of the last layer as the output of the network. Combined with the output of high and low levels, the equipment degradation process can be learned more comprehensively. The implementation process of the multi-scale convolutional neural network is shown in the MSCNN part of Figure 3. In this part, the output of the last pooling layer in the CNN and the output of the last two convolutional layers are connected and flattened as the final output of the CNN. The hyperparameters of the MSCNN are shown in Table 1.

### 3.2. Bidirectional Long Short-Term Memory Network (BiLSTM)

The BiLSTM model is a time-recurrent neural network, which is composed of forward LSTM and reverse LSTM [23]. The signal is, respectively, input into two LSTM neural networks in positive and reverse order for feature extraction, and the two output vectors (that is, the extracted feature vectors) are spliced to form a vector as the final feature output [24,25]. The feature data obtained by the BiLSTM model at the t moment simultaneously have past and future information. The neural network structure model is superior to the single LSTM structure model in feature extraction efficiency and performance. The BiLSTM structure is shown in Figure 4.

The data are entered into BiLSTM through the input layer. A value is obtained by LSTM forward calculation, and a value is also obtained by LSTM backward calculation. The value in the hidden layer is determined by these two values, and the formulae are as follows:(2)$$\vec{h_{t}}=f(\vec{w_{x}}\cdot X_{t}+\vec{h_{t-1}}\cdot \vec{w_{h}}+\vec{b_{n}}),$$
(3)$$\overset{\leftarrow }{h_{t}}=f(\overset{\leftarrow }{w_{x}}\cdot X_{t}+\overset{\leftarrow }{h_{t-1}}\cdot \overset{\leftarrow }{w_{h}}+\overset{\leftarrow }{b_{n}}),$$
(4)$$Y_{t}=f(\vec{w_{y}}\cdot \vec{h_{t}}+\overset{\leftarrow }{w_{y}}\cdot \overset{\leftarrow }{h_{t}}+b_{y}),$$
where f is the activation function, w is the weight and bias term, $\vec{h_{t}}$ is the output of the front layer, $\overset{\leftarrow }{h_{t}}$ is the output of the back layer and Yt is the final output.

## 4. SAF Experiment under Vibration Conditions

### 4.1. Experimental Device Design

In order to detect the SAF under different vibration conditions, the SAF experimental platform is designed, as shown in Figure 5, and the experimental circuit diagram is shown in Figure 6. The power supply of the experimental platform is AC 380V, and the load is a three-phase asynchronous motor Y160M-6-11 KW and frequency converter VFD110E43A. In order to better simulate the arc fault caused by the mechanical vibration of the contact point caused by the motor fault, the arc fault generator is placed between the frequency converter and the motor, and the loop current is adjusted by the magnetic powder brake during the experiment. The current sensor LHB100A5VY2 collects the fault phase current at the front end of the frequency converter, and the voltage sensor LHB500V5VT1 is used to detect the arc fault voltage at the back end of the frequency converter. The data acquisition card USB3200 transmits the current signals to the computer, and completes the data storage and display in Labview2014.

The arc fault generator is shown in Figure 7. The arc fault generator, respectively, uses a cylindrical carbon rod and a conical copper rod with a diameter of 5 mm as the electrode. In order to simulate the different vibration conditions caused by the motor fault, the OWS80-06-LB-05 voice coil motor is used to control the moving electrode to move with different vibration frequencies and amplitudes. The arc fault generated by the vibration of the electrical contact point caused by the motor fault is simulated. The voice coil motor is controlled by an ADP-090-09 driver, and the driver power supply is an LRS-100-24 switching power supply.

### 4.2. Experimental Scheme Design

According to Reference [26], the vibration frequency of the three-phase asynchronous motor Y160M-6-11 KW used in this paper is 16.2 Hz when mechanical faults, such as rotor imbalances, air gap dynamic eccentricities and the poor processing and assembly of rolling bearings, occur. When mechanical looseness and other faults occur, the vibration frequency is 25 Hz, as shown in Table 2. In order to better study the influence of mechanical vibration on the SAF, this study designs the experimental scheme of the motor under different working conditions, which are normal operation, no vibration arc fault, 16 Hz vibration arc fault and 25 Hz vibration arc fault. The loop current and power supply voltage in the experiment are 12 A and 380 V, respectively, and the sampling frequency is 50 KHz. The specific experimental scheme is shown in Table 3.

### 4.3. Analysis of Experimental Results

The current waveform during normal operation is shown as Figure 8a, and the current waveforms at the front end of the frequency converter when the arc fault occurs at the back end of the frequency converter are shown as Figure 8b–d. Although the frequency converter is full-wave rectified, the output filter capacitor will cause the input side current to be a double-peak pulse waveform. It can be found from Figure 8b,d that, when the arc fault occurs in the line, the peak current in the half-cycle wave increases, and some peaks disappear. This is because the electrode separation cannot pull out the arc if it happens to be in a long arc zero-current area (LAZCA), resulting in the disappearance of some peaks. By comparing Figure 8c,d, it can be found that the smaller the vibration frequency, the longer the electrode separation in the arc-free state, and the higher the number of disappearing peaks.

The arc voltage waveform is shown in Figure 9. It can be seen from this figure that, when the arc fault occurs, the saddle shape appears in the voltage waveform. By comparing Figure 9b,c, it can be seen, that under vibration conditions, there is a stable saddle shape and an irregular saddle shape. This is because under vibration conditions, due to the vibration of the contact point, the arc fault fluctuates continuously between the states of ignition and extinction, resulting in a continuous change in the arc resistance. The arc fault under no vibration continues to burn, and the arc resistance is relatively stable. In addition, by comparing Figure 9c,d, it can be seen that, with the increase in vibration frequency, the arc fault voltage waveform does not change significantly. Therefore, it is necessary to further process and analyze the voltage data to realize the detection of the SAF.

## 5. SAF Experiment under Vibration Conditions

### 5.1. Feature Extraction Results

The decomposition results of arc current and arc voltage by NA-MEMD are shown in Figure 10. After processing, the SAF signal is decomposed into 10 IMFs. As the order of the decomposed IMFs increases, the frequency range it contains becomes larger and larger, that is, more and more fault information is contained [27]. The correlation coefficients of the IMF components of the current and voltage decomposed by NA-MEMD are shown in Table 4 and Table 5. From Table 4, it can be seen that the correlation coefficients of IMF7–IMF10 are much larger than that of IMF1–IMF6 in the decomposed current IMF components. The higher the correlation coefficient, the stronger the relationship between the IMF component and the original signal. Therefore, in order to form a new arc fault current characteristic signal, this paper selects IMF7–IMF10, four sets of data for data reconstruction. However, in the voltage data, the correlation coefficients of IMF8–IMF10 are larger than that of IMF1–IMF7. Therefore, for the arc voltage data, IMF8–IMF10 are selected for data reconstruction to form a new voltage characteristic signal.

### 5.2. Real-Time Comparison of Data Feature Extraction Methods

In this paper, white Gaussian noise is added to the data feature extraction many times, which increases the number of calculations. And the arc fault detection has high real-time requirements [28]. In order to analyze the real-time performance of the algorithm in this paper, the current data under different vibration frequencies are input into different feature extraction methods for feature extraction, and the fault features extracted by different algorithms are input into the MSCNN-BiLSTM neural network for training. The test results are shown in Table 6.

It can be seen from Table 6 that the real-time performance of the data feature extraction method proposed in this paper is significantly better than that of CEEMDAN. Although the real-time performance of NA-MEMD is not as good as the original EMD, wavelet decomposition and other data feature extraction methods, the accuracy of the detection model of data feature extraction through NA-MEMD is much better than the original EMD and wavelet decomposition. Considered comprehensively, the data feature extraction algorithm based on NA-MEMD proposed in this paper is more suitable for extracting the features of the SAF under different vibration frequencies.

### 5.3. Arc Fault Detection Results and Analysis

The SAF current and voltage data processed by NA-MEMD are classified into four categories: normal, 16 Hz vibration arc fault, 25 Hz vibration arc fault and no vibration arc fault. Each fault contains 800 samples of current and voltage data, with a single sample length of 1250 time series points. Among them, 640 groups are used to train the model, and the other 160 groups are used to detect the accuracy of MSCNN-BiLSTM in identifying arc fault current data. The final classification results are shown in Figure 11. It can be seen from Figure 11 that the SAF detection method proposed in this paper cannot only realize the detection of the SAF, but can also distinguish the vibration frequency of the arc fault, so as to preliminarily judge the cause of the mechanical fault in the motor.

In order to verify the superiority of the classification effect of the NA-MEMD-MSCNN-BiLSTM (N-M-M-B) algorithm proposed in this paper, MSCNN-BiLSTM is used to classify the arc fault current data separately, and the number of samples and the network structure remain unchanged. From the comparison of the two confusion matrices in Figure 11 and Figure 12, it can be seen that, compared with the analysis of single current data, the classification result of the current and voltage data as features at the same time is more accurate, and the accuracy of fault detection is also improved.

It can also be seen from Figure 11 and Figure 12 that the model proposed in this paper can better realize the classification of the vibration frequency of the SAF under vibration conditions. According to the data in Table 5, it can be seen that, at the vibration frequency of 16 Hz, the mechanical faults usually caused by the Y160M-6-11 KW motor are rotor imbalance, air gap dynamic eccentricity, and the poor processing and assembly of rolling bearings. When mechanical looseness and other faults occur, the vibration frequency is usually 25 Hz. Thus, we can realize the preliminary judgment of the motor fault type according to the classification results.

### 5.4. Comparison of Algorithm Performance

#### 5.4.1. Comparison and Analysis with Other Detection Methods

In order to verify the effectiveness of the proposed algorithm, the proposed algorithm is compared with other methods in the field of motor frequency converter load arc fault detection. The comparison results are shown in Table 7. It can be seen from Table 7 that the algorithm proposed in this paper is more applicable to the problem of arc fault at the back end of the frequency converter caused by a mechanical fault in the motor. Compared with MSCNN-LSTM, MSCNN-BiLSTM can not only accurately detect the arc fault, but also has a better multi-classification effect.

#### 5.4.2. Noise Immunity Analysis of the Model

In order to verify the noise immunity of the proposed algorithm, 10 dB, 20 dB and 30 dB white Gaussian noise signals are input into the current signals, and the noise immunity of N-M-M-B is analyzed. The results are shown in Table 8. It can be seen from Table 8 that the model can achieve a higher accuracy under the conditions of 10 dB, 20 dB and 30 dB noise signal interference, which shows that the model has better noise immunity.

#### 5.4.3. Generalization Analysis of the Model

N-M-M-B is used to detect the arc fault of the motor load without a frequency converter and with no vibration arc fault of the front end with a frequency converter load. The detection results are shown in Table 9. It can be seen from the experimental results that the algorithm proposed in this paper can achieve better detection for the motor load without a frequency converter and the front-end motor load of a frequency converter, and has higher detection accuracy. Therefore, the algorithm proposed in this paper has good generalization.

In summary, the N-M-M-B proposed in this paper not only has high detection accuracy for SAF detection, but also has good noise immunity and generalization.

## 6. Conclusions

In this paper, the back-end arc fault experiment of a frequency converter under vibration conditions was carried out. The SAF generated by the electrical contact point at a specific vibration frequency under the mechanical fault state of the motor was simulated. Compared with the fault simulation at the front-end of the frequency converter, the scheme proposed in this paper is more suitable for an actual SAF under the mechanical vibration of a motor. In this paper, noise-assisted multivariate empirical mode decomposition is used to extract the features of arc current and voltage, which reduces the time of feature extraction. Compared with CEEMDAN and other algorithms with high detection accuracy, the feature extraction time of a single sample is shorter, which reduces the calculation amount for subsequent detection and improves the detection accuracy.

Finally, this paper proposes an NA-MEMD-MSCNN-BiLSTM model based on SAF detection. The accuracy of this model for SAF detection at the back-end of a frequency converter can reach 99.69%. Compared with the detection methods in other studies, this study not only improves the detection accuracy, but also controls the detection time of a single sample at 0.044 ms. While improving the detection accuracy, the detection time is controlled, which provides the theoretical basis for the development of subsequent arc fault detection devices and realizes the preliminary judgment of the cause of motor mechanical faults. In addition, the detection accuracy of the model proposed in this paper can reach 99.37%, 98.75% and 95.94%, respectively, under 10 dB, 20 dB and 30 dB noise signal interference, and it still has high detection accuracy under different fault conditions. Therefore, the model proposed in this paper has good noise immunity and generalization.

## 7. Future Work Prospects

In this paper, due to the limitations of the experimental conditions, we did not design an actual arc fault detection device. In future work, we plan to use Micro Control Unit (MCU) to design an arc fault detection device. By embedding the trained detection algorithm into MCU and inputting the real-time generated data into MCU, the real-time detection of the SAF will be realized.

## Figures and Tables

**Figure 1 sensors-24-00959-f001:**
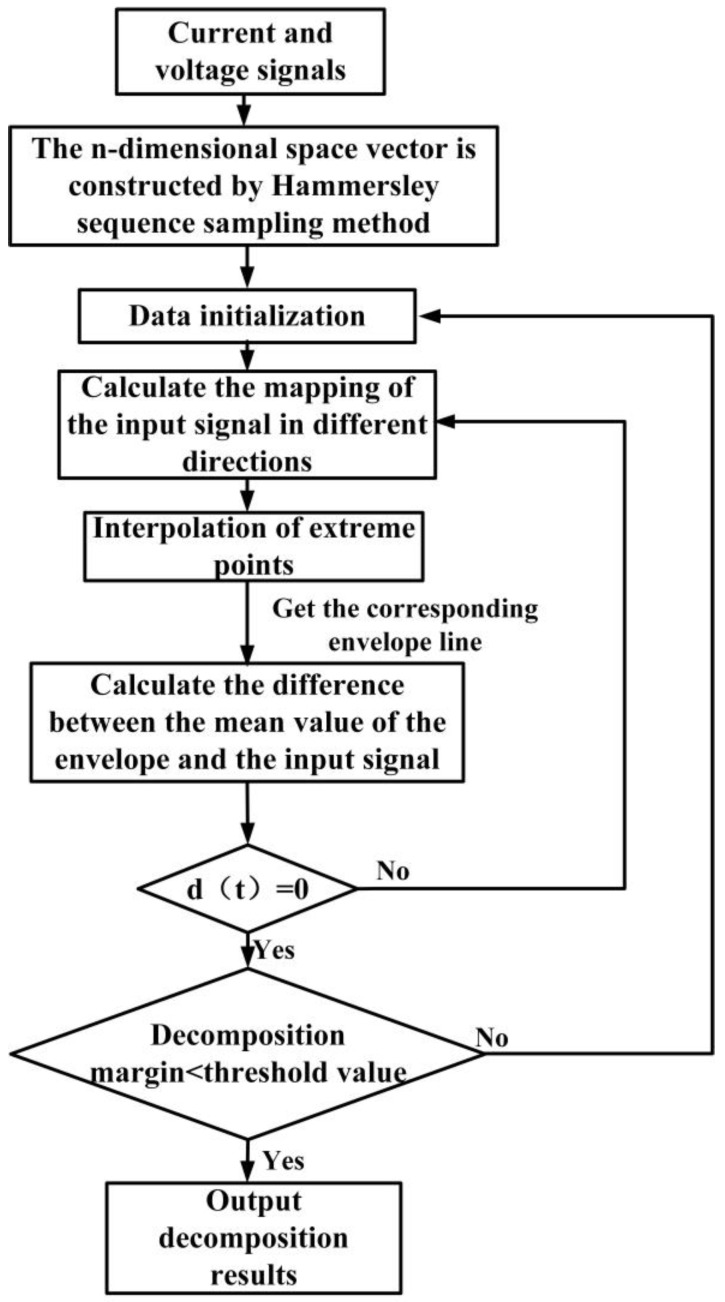
MEMD algorithm flow chart.

**Figure 2 sensors-24-00959-f002:**
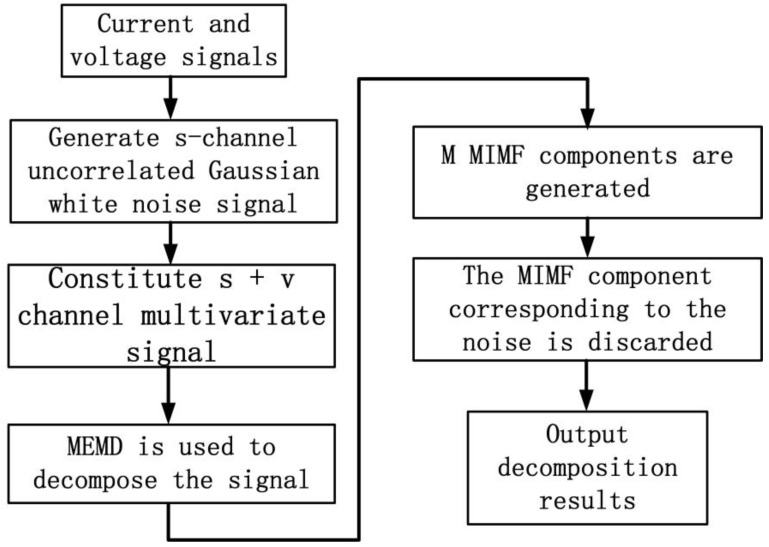
NA-MEMD algorithm flow chart.

**Figure 3 sensors-24-00959-f003:**
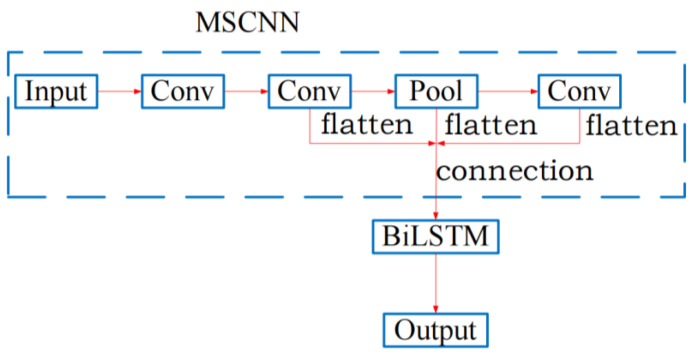
Structure diagram of the MSCNN-BiLSTM network.

**Figure 4 sensors-24-00959-f004:**
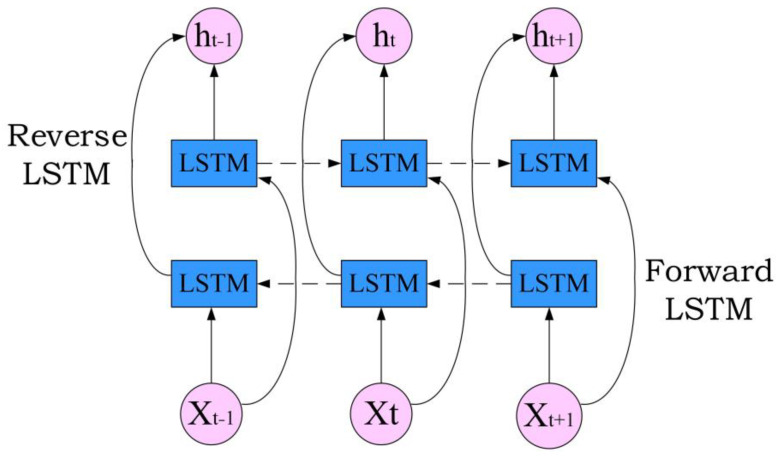
BiLSTM structure diagram.

**Figure 5 sensors-24-00959-f005:**
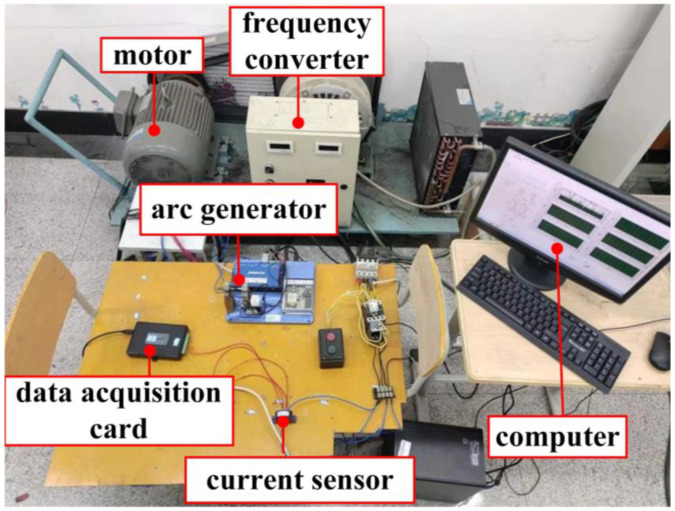
Experimental platform.

**Figure 6 sensors-24-00959-f006:**
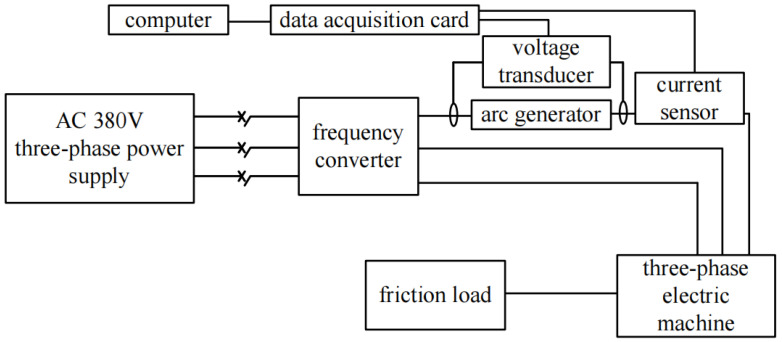
Experimental circuit diagram.

**Figure 7 sensors-24-00959-f007:**
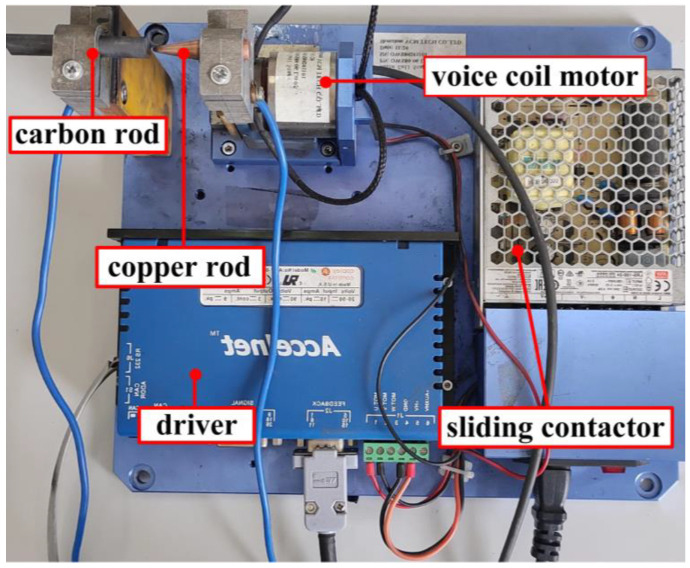
Arc fault generator.

**Figure 8 sensors-24-00959-f008:**
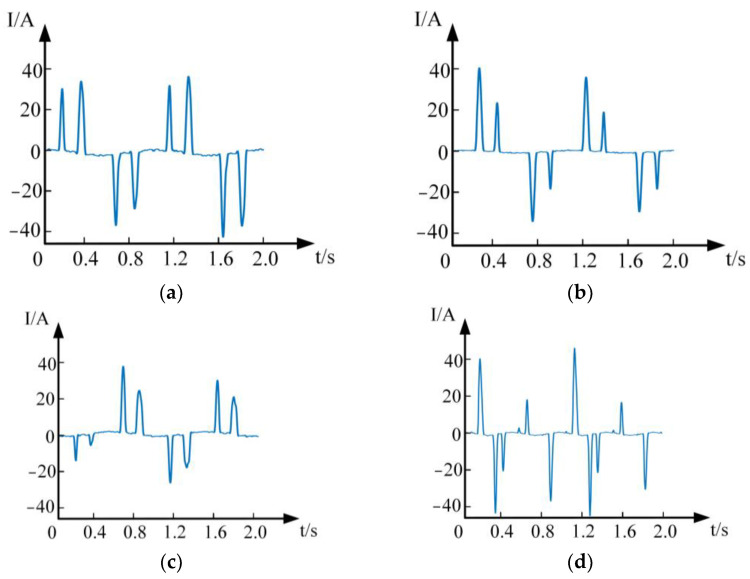
Experimental current waveform. (**a**) 12 A/normal. (**b**) 12 A/no vibration arc fault. (**c**) 12 A/16 Hz vibration arc fault. (**d**) 12 A/25 Hz vibration arc fault.

**Figure 9 sensors-24-00959-f009:**
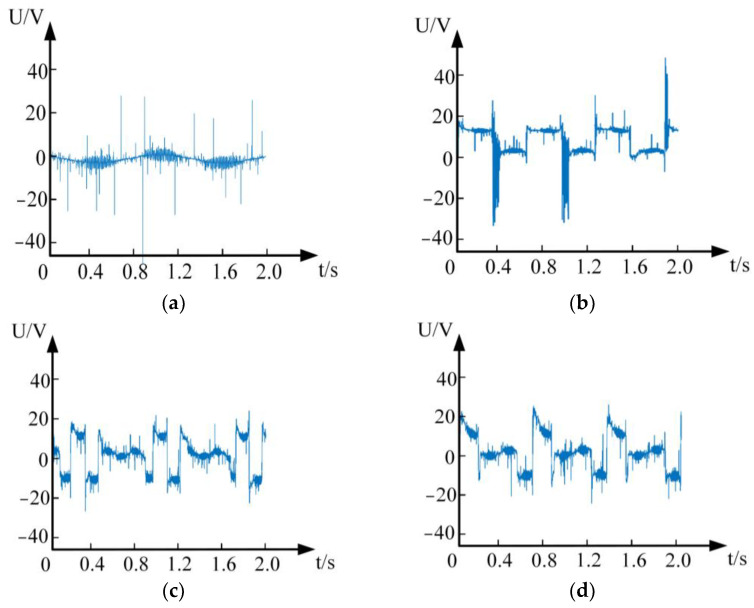
Experimental voltage waveform. (**a**) 12 A/normal. (**b**) 12 A/no vibration arc fault. (**c**) 12 A/16 Hz vibration arc fault. (**d**) 12 A/25 Hz vibration arc fault.

**Figure 10 sensors-24-00959-f010:**
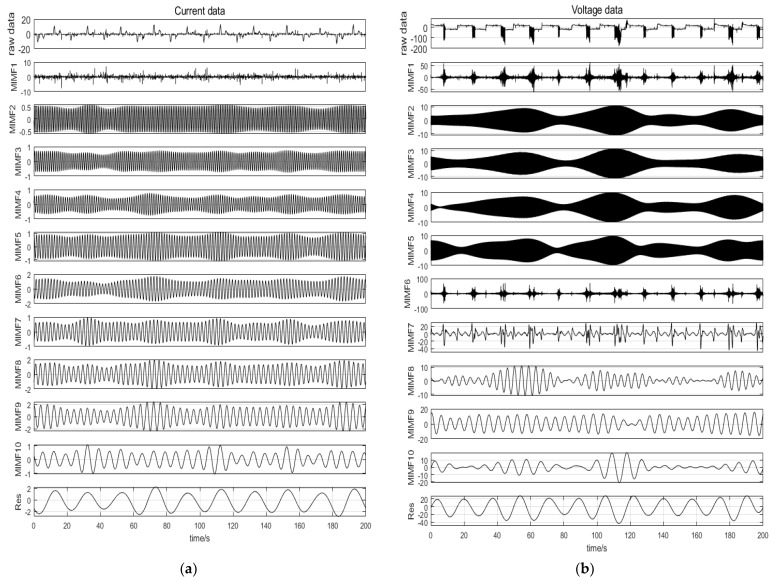
Typical IMFs decomposed by NA-MEMD algorithm. (**a**) Decomposition results of 12 A/16 Hz arc fault current data. (**b**) Decomposition results of 12 A/16 Hz arc fault voltage data.

**Figure 11 sensors-24-00959-f011:**
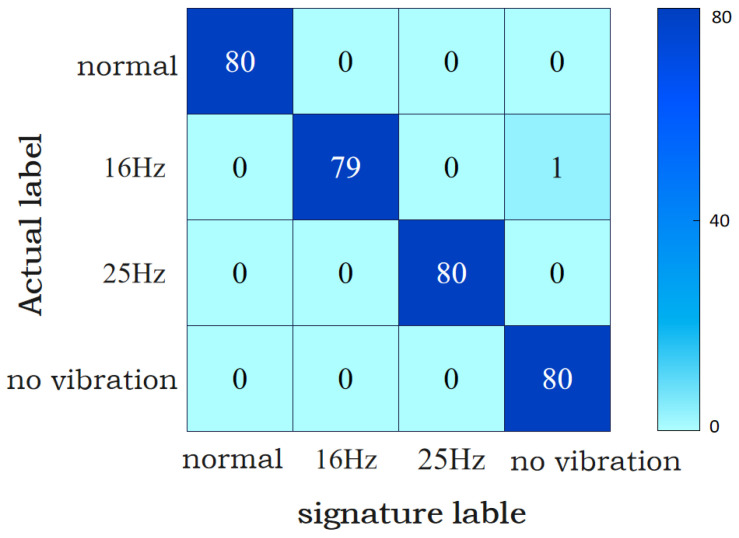
N-M-M-B classification results.

**Figure 12 sensors-24-00959-f012:**
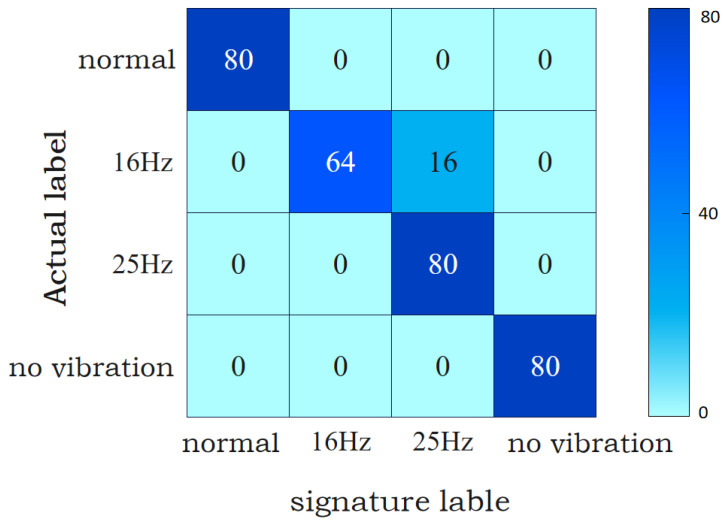
MSCNN-BiLSTM classification results.

**Table 1 sensors-24-00959-t001:** MSCNN network hyperparameters.

No.	Neural Network Layer	Hyperparameter, Output Channel Size, Activation Function Selection
1	Input layer	Size = 128 × 128 × 1; channel = 1
2	Convolution layer 1	Size = 3, strides = 2; filters = 50, kernel = 20; relu
3	Convolution layer 2	Size = 3, strides = 2; filters = 30, kernel = 10; relu
4	Maximum pooling layer	Size = 2, strides = 2; data_format = channels_last
5	Convolution layer 3	Size = 3, strides = 1; filters = 30, kernel = 6; relu

**Table 2 sensors-24-00959-t002:** Main motor faults and frequencies.

Motor Fault Type	Vibration Frequency (Hz)
Rotor imbalance, air gap dynamic eccentricity, poor processing and assembly of rolling bearings	16.2
Mechanical looseness	25

**Table 3 sensors-24-00959-t003:** Experimental scheme.

Group NO.	Vibration Frequency (Hz)	Vibration Amplitude (mm)	SAF
1	0	0	No
2	0	0	Yes
3	16	55	Yes
4	25	55	Yes

**Table 4 sensors-24-00959-t004:** IMF correlation coefficients of 12 A/16 Hz arc fault current data decomposed by NA-MEMD algorithm.

IMF	Correlation Coefficient	IMF	Correlation Coefficient
IMF1	0.0177	IMF6	0.0284
IMF2	0.0196	IMF7	0.4372
IMF3	0.0318	IMF8	0.4904
IMF4	0.0210	IMF9	0.7041
IMF5	0.0239	IMF10	0.5868

**Table 5 sensors-24-00959-t005:** IMF correlation coefficients of 12 A/16 Hz arc fault voltage data decomposed by NA-MEMD algorithm.

IMF	Correlation Coefficient	IMF	Correlation Coefficient
IMF1	0.0199	IMF6	0.0786
IMF2	0.0260	IMF7	0.0832
IMF3	0.0255	IMF8	0.1059
IMF4	0.0459	IMF9	0.1033
IMF5	0.0284	IMF10	0.4703

**Table 6 sensors-24-00959-t006:** Test results of different feature extraction methods.

Feature Extraction Method	Accuracy	Single Sample Feature Extraction Time (ms)
KPCA	76.99%	63.44
Wavelet Decomposition	68.26%	20.06
CEEMDAN	98.71%	93.11
EMD	76.33%	55.71
EEMD	89.11%	72.624
Proposed algorithm (NA-MEMD)	99.69%	65.46

**Table 7 sensors-24-00959-t007:** Fault detection accuracy of different algorithms.

Algorithm	Accuracy	Detection Time of Model for a Single Sample (ms)
Reference [9]	88.18%	0.071
Reference [10]	80.36%	0.038
Reference [12]	81.49%	0.16
Reference [14]	90.16%	0.087
MSCNN-LSTM	93.12%	0.059
Proposed algorithm (N-M-M-B)	99.69%	0.044

**Table 8 sensors-24-00959-t008:** Noise immunity verification.

Signal-to-Noise Ratio (SNR)/dB	Accuracy
30	99.37%
20	98.75%
10	95.94%

**Table 9 sensors-24-00959-t009:** Detection results of the SAF data with different loads.

Data Type	Accuracy
Arc current data of the motor load without frequency converter	99.06%
No vibration arc current data of the front end with frequency converter load	98.12%

## Data Availability

Due to the nature of this research, the data presented in this study are available on request from the corresponding author.

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
