# Peer review of "Series Arc Fault Detection under Vibration Condition Based on N-M-M-B"

_sensors, 2024, doi:10.3390/s24030959_

Round 1
Reviewer 1 Report
Comments and Suggestions for Authors
After reviewing the manuscript, here are my comments:
1- Please highlight the novelty of the proposed method in comparison to the other methods in the field.
2- Please increase quality of the figures.
3- Please reorganize the conclusion and enrich it by data.
Author Response
Response to Reviewer X Comments
|
||
1. Summary |
|
|
Thank you very much for reviewing our manuscripts in your busy time. By carefully reading your review opinions, we found our own shortcomings and carefully revised the paper. In this process, we benefited a lot. Thank you again for your valuable comments on the paper.
|
||
2. Questions for General Evaluation |
Reviewer’s Evaluation |
Response and Revisions |
Does the introduction provide sufficient background and include all relevant references? |
Can be improved |
We have added references to the article |
Are all the cited references relevant to the research? |
Yes |
|
Is the research design appropriate? |
Yes |
|
Are the methods adequately described? |
Can be improved |
We have modified the description of the content of the article in the introduction in order to clearly express the research content of the article. |
Are the results clearly presented? |
Can be improved |
We can modify the conclusion of the article so that we can clearly express the conclusion of the article. |
Are the conclusions supported by the results? |
Can be improved |
We add data description to the conclusion so that the conclusion can be fully supported by the data. |
3. Point-by-point response to Comments and Suggestions for Authors |
||
1、Comments 1: Please highlight the novelty of the proposed method in comparison to the other methods in the field. |
||
Response 1: Thanks for your suggestions, we have added a section in the introduction to compare with other algorithms in this field, so as to emphasize the novelty of the algorithm proposed in this paper. The details are as follows : To address the above problems, an arc fault generator with adjustable vibration frequency and amplitude is designed in this paper, which can simulate the SAF under different vibration conditions. The noise-assisted multivariate empirical mode decomposition (NA-MEMD) arc feature extraction method is constructed by combining the noise channel signal and the multivariate empirical mode decomposition, so as to better extract the frequency domain characteristics of the SAF current under vibration condition and solve the problem of feature extraction modal aliasing. In the meanwhile, the arc voltage and circuit are simultaneously analyzed to avoid misjudgment caused by single information. Finally, the multi-scale technology is combined with the traditional CNN and bidirectional long short-term memory network (BiLSTM) to construct MSCNN-BiLSTM arc fault detection model, so that BiLSTM module can receive more and more abundant multi-level features from CNN and improve the accuracy of the detection model. |
||
Comments 2: Please increase quality of the figures. |
||
Response 2: Thank you for your comments. We are very sorry that the figures in our paper do not meet the clarity requirements. We have modified the unclear figures in the paper to improve the quality of the figures.
|
||
Comments 3:Please reorganize the conclusion and enrich it by data. |
||
Response 3: Thank you for your comments on the paper, which gives us a deeper understanding of how academic contributions and innovations should be condensed. We have modified the conclusion part and enriched it by using the data. The results are as follows : 6.Conclusion In this paper, the back-end arc fault experiment of frequency converter under vibration condition is carried out. The SAF generated by the electrical contact point at a specific vibration frequency under the mechanical fault state of the motor is simulated. Compared with the fault simulation at the front-end of the frequency converter, the scheme proposed in this paper is more suitable for the actual SAF under the mechanical vibration of the motor. In this paper, noise-assisted multivariate empirical mode decomposition is used to extract the features of arc current and voltage, which reduces the time of feature extraction. Compared with CEEMDAN and other algorithms with high detection accuracy, the feature extraction time of single sample is shorter, which reduces the calculation amount for subsequent detection and improves the detection accuracy. Finally, this paper proposes a CEEMDAN-MPE-MSCNN-BiLSTM model based on the SAF detection. The accuracy of this model for SAF detection at the back-end of frequency converter can reach 99.69 %. Compared with the detection methods in other papers, this paper not only improves the detection accuracy, but also controls the detection time of a single sample at 0.044 ms. While improving the detection accuracy, the detection time is controlled, which provides the theoretical basis for the development of subsequent arc fault detection devices and realizes the preliminary judgment of the cause of motor mechanical faults. In addition, the detection accuracy of the model proposed in this paper can reach 99.37 %, 98.75 % and 95.94 % respectively under 10dB, 20dB and 30dB noise signal interference, and it still has high detection accuracy under different fault conditions. Therefore, the model proposed in this paper has good noise immunity and generalization.
|
Reviewer 2 Report
Comments and Suggestions for Authors
The paper covers important topic. The title of the paper contains abbreviation N-M-M-B which is not well known therefore it is preferable to change the title. Experimental section it is better to put after theoretical explanation. The electrical circuit of the setup should be shown. In the paper more graphical information could be useful to explain idea. Some pictures have poor quality. The paper has format mistakes as well. Data used for the training of AI algorithm should be explained more. Practical implementation should be rewield more detailed.
Reviewer 3 Report
Comments and Suggestions for Authors
In my opinion, your work is original, you have described in it the research carried out correctly methodologically. You have demonstrated your own contribution to the development of the field. Also, carefully you conducted a state of the art survey. In your work I did not notice the features of: plagiarism, excessive self-citation or other ethical violations. Therefore, in my opinion, your work can be published after a slight correction. At the same time, after the corrections, your work will not require a re-review. My comments do not concern the scientific or metological aspects, but the editing side.
It is worth changing the layout of the work a little, to more clearly separate the description of your method and the network used - from other solutions used. And so the description of the state of the art is mixed in the paper with the description of your method and research results.
The results obtained need to be described in more detail, a description of the conduct of tests can be added, the number of tests carried out can be given. Finally, what I miss is a description of how to implement your method under industrial conditions. Each method/idea is as good as it is useful. Sometimes it is difficult to put a computer next to an engine in an industrial setting.
Some specific comments:
- we should avoid defining abbreviations in the Abstract,
- there are too short/single-sentence paragraphs in the paper - e.g. on p. 2,
- don't really understand the purpose of numbering paragraphs - as on p. 3,
- there should be a space before Figure 2,
- title of subsection 2.3. move to the next page,
- no blank space after the captions of Figures 3 and 5-7 and 9,
- figure 4 can not be placed on two pages,
- caption under figure 5 on the same page as the figure,
- description of activities under figure 5 is worth improving - to make the text easier to read,
- line 218 does not contain the design number.
Reviewer 4 Report
Comments and Suggestions for Authors
Series Arc Fault Detection Under Vibration Condition Based on N-M-M-B
This is an interesting paper. In order to improve the paper quality, the authors are required to take the following notes.
1 1. The paper title needs to be revised as nobody understands the meaning of N-M-M-B. You can attempt to use the abbreviations of those neural networks tested in this paper.
2 2. Please note any Acronym must be defined before its first use.
3 3. In Table 2, why do you use Group 1 and Group 2 without vibration frequency and vibration magnitude?
4 4. Please explain why the correlation coefficient values are so low for IMF7 ~ IMF9 in Table 4?
5 5. Explain what the fault features are in Section 3.3.
6 6. Throughout the text, you discussed the fault detection but not the fault diagnosis. Therefore, change diagnosis to detection in Section 4.3.5.
7 7. In Conclusion, there is one sentence as “The model can not only distinguish normal, no vibration arc and arc fault under different vibration frequencies, but also preliminarily judge the cause of mechanical fault of the motor.” Please make sure that you have judged the causes of mechanical faults in the text before the conclusion section.
Comments on the Quality of English LanguageSome obvious English grammar errors are identified. Please check the text carefully and remove these errors.
Round 2
Reviewer 2 Report
Comments and Suggestions for Authors
The paper has been improved.